# Solid Lipid Nanoparticles: Review of the Current Research on Encapsulation and Delivery Systems for Active and Antioxidant Compounds

**DOI:** 10.3390/antiox12030633

**Published:** 2023-03-03

**Authors:** Edy Subroto, Robi Andoyo, Rossi Indiarto

**Affiliations:** Department of Food Industrial Technology, Faculty of Agro-Industrial Technology, Universitas Padjadjaran, Bandung 45363, Indonesia

**Keywords:** nanoencapsulation, solid lipid nanoparticle, active compound, antioxidant, delivery system

## Abstract

Various active compounds are easily damaged, so they need protection and must be easily absorbed and targeted. This problem can be overcome by encapsulating in the form of solid lipid nanoparticles (SLNs). Initially, SLNs were widely used to encapsulate hydrophobic (non-polar) active compounds because of their matched affinity and interactions. Currently, SLNs are being widely used for the encapsulation of hydrophilic (polar) and semipolar active compounds, but there are challenges, including increasing their entrapment efficiency. This review provides information on current research on SLNs for encapsulation and delivery systems for active and antioxidant compounds, which includes various synthesis methods and applications of SLNs in various fields of utilization. SLNs can be developed starting from the selection of solid lipid matrices, emulsifiers/surfactants, types of active compounds or antioxidants, synthesis methods, and their applications or utilization. The type of lipid used determines crystal formation, control of active compound release, and encapsulation efficiency. Various methods can be used in the SLN fabrication of active compounds and hydrophilic/hydrophobic antioxidants, which have advantages and disadvantages. Fabrication design, which includes the selection of lipid matrices, surfactants, and fabrication methods, determines the characteristics of SLNs. High-shear homogenization combined with ultrasonication is the recommended method and has been widely used because of the ease of preparation and good results. Appropriate fabrication design can produce SLNs with stable active compounds and antioxidants that become suitable encapsulation systems for various applications or uses.

## 1. Introduction

Utilization of lipid-based nanotechnology for encapsulation and delivery systems for active compounds can be applied in several models, including nanoliposomes, nanosuspensions, nanoemulsions, solid lipid nanoparticles (SLNs), and nanostructured lipid carriers (NLCs) [1,2,3,4,5]. SLNs are the most popular models because they can provide solid and stable nanoparticles, have a good delivery system, and can be modified with various lipid matrices, emulsifiers, core materials, and fabrication methods. SLNs were introduced in 1991 and are a better carrier system than conventional colloid systems [6,7,8]. SLNs have become an effective and stable model for encapsulation and delivery systems for liposomes, emulsions, and polymer nanoparticles mainly because of their ease of forming dispersion systems and adequate entrapment efficiency [9]. SLNs are colloidal carrier systems consisting of a lipid core with a high melting point and are coated with a liquid surfactant, where a surfactant with a concentration of about 0.5–5% can increase the dispersion stability of SLNs [8,10,11].

SLNs consist of main ingredients, namely solid lipids, surfactants/emulsifiers, and water or other solvents. The lipid used as the dispersed phase is a solid lipid that serves as a matrix material for the introduction of encapsulated compounds [12]. The types of solid lipids can be in the form of triglycerides, partial glycerides, free fatty acids, steroids, and waxes [4,13]. The phase change of liquid lipids to solid lipids has new benefits for SLN colloidal carrier systems, because it can improve control of the release kinetics of coated compounds and increase the stability of the formed lipophilic sites. The use of solid lipids can improve the control of the release of micronutrients trapped in lipids in the gastrointestinal system, compared to liquid lipids, and can maintain the stability of micronutrients from environmental conditions such as water, light, and oxygen [14,15,16]. In addition, the dense lipid matrix can slow the digestion of lipids, thereby allowing the release of the encapsulated compounds in a more sustainable or long-lasting manner. While in the emulsifying phase, the surfactant used is a liquid-type surfactant in order to form an O/W emulsion or the double emulsion (W/O/W), and at the same time, it can function as a stabilizer for SLN dispersion [17,18].

The selection of lipids and surfactants in the SLN synthesis/fabrication process affects the characteristics of SLNs, such as entrapment efficiency, particle size, stability, crystallinity, and pharmacokinetic properties such as bioavailability, release time, and absorption stability of the coated drug or active ingredient [19,20]. The methods that can be used for the fabrication of SLNs include high-speed homogenization combined with ultrasonication, high-pressure homogenization at high or low temperatures, evaporation or diffusion, solvent emulsification, and supercritical fluid extraction of emulsions [10,21]. The method for the fabrication of the selected SLNs must be adapted to the characteristics of the active ingredient to be encapsulated, so that the application is suitable for the objectives to be achieved [22,23].

The active ingredients encapsulated by SLNs include drugs, vitamins, minerals, antibiotics, and various polar, semipolar, and non-polar antioxidant compounds. The development of SLNs in antioxidant compounds has attracted attention, because SLNs are very effective in increasing antioxidant activity, bioavailability, and controlled release for absorption in the gastrointestinal system [24,25,26]. Various antioxidant compounds have been successfully encapsulated in SLNs using various methods, lipid matrices, and emulsifiers or surfactants. Hydrophobic antioxidant compounds are relatively easy to make into SLNs because of their affinity and suitable interactions between the lipid matrix and the core material [10,27,28,29]. Meanwhile, hydrophilic antioxidants require certain treatments to be encapsulated well, especially through the formation of emulsions, both single emulsion and double emulsion [30,31,32].

Antioxidant-loaded SLNs are becoming more of an object of interest, along with the increasing need for well-protected antioxidant compounds to improve the stability of foodstuffs and increasing interest by the pharmaceutical industry, due to their quality as a delivery system for optimal absorption [11,33,34]. In addition, specific discussions regarding the application of SLNs for antioxidant compounds are still limited. Therefore, this review aims to provide information on current research on SLNs, starting with an overview of various fabrication methods and their applications, especially for the encapsulation and delivery systems for various active compounds, especially antioxidant compounds, both non-polar (hydrophobic) and polar (hydrophilic) antioxidants.

## 2. Development of Lipid Nanoparticles Technology in the Form of Solid Lipid Nanoparticles (SLNs) and Nanostructured Lipid Carriers (NLCs)

Lipid nanoparticle formulation (LNF) as a delivery system provides many benefits related to the encapsulation of drugs and active compounds, providing kinetic stability and rigid structural morphology compared to other lipid-based vehicles, such as liposomes and emulsions in colloidal nanocarrier systems. Various kinds of LNFs can be developed, especially in the form of SLNs and NLCs [10,35,36,37,38,39]. SLNs are delivery systems with a solid lipid fraction to increase the bioavailability of micronutrients or active compounds that have low solubility [40,41,42]. SLNs are very good carriers for colloidal particles, such as emulsions, liposomes, micro-polymers, and nanoparticles [8,43]. SLNs consist of three main ingredients, namely active components or drugs, solid lipids, and surfactants. Surfactants act as a barrier between the lipid matrix on the outside of the particles. Solid lipids that can be used are saturated fats in the form of free fatty acids, mono- and diglycerides, triglycerides, and other fatty acid esters that act as binders of active substances or drugs in the lipid matrix [43,44,45].

SLNs are composed of a dense lipid matrix at physiological temperatures, surfactants, and sometimes added cosurfactants (such as solvents) [10,19,46]. SLNs can be synthesized by various methods, such as solvent emulsification/evaporation, cold homogenization, hot homogenization, high-pressure homogenization, high-shear homogenization, and ultrasound methods [39]. The synthesis/fabrication of SLNs must optimize the mixture of lipids used with surfactants and other materials, so that the resulting mixture can form a thermodynamically stable and transparent microemulsion [23,47]. However, the temperature used and preparation conditions can affect the functionality of SLNs. The use of high temperatures can also reduce the stability of the active compounds encapsulated in SLNs. In addition, the use of high temperatures during the preparation of SLNs and rapid cooling during solidification can lead to the formation of unstable lipid crystals and affect the entrapment efficiency [35,48].

The selection of appropriate lipids and surfactants can also affect micronutrient loading, stability during storage, particle size, and release behavior. The lipids used are biocompatible (e.g., glyceryl monostearate, tripalmitin, stearic acid, cetyl alcohol, Compritol 888, tristearin, etc.), in solid form at room temperature and body temperature. Surfactants that function as emulsifiers must be able to reduce the surface tension between the two phases, so that the hydrophilic groups as the head and the lipophilic groups that form the tail of the surfactant must have a suitable value of hydrophilic-lipophilic balance (HLB) [21]. SLNs, as micronutrient delivery systems, coat bioactive components, vitamins, minerals, and other active compounds to prevent these micronutrients from being damaged on the way to the absorption process in the intestine. SLNs in food products are generally used as a form of fortification with micronutrient components needed by the body [16,49].

### 2.1. The Superiority and Weakness of SLNs

SLNs are an emulsion system in the form of lipid nanoparticles, which was developed as an alternative carrier system that replaces previous methods such as emulsions, liposomes, and polymer nanoparticles [6,8,50]. SLNs are encapsulation and delivery systems that have been widely developed, because they have various advantages in various respects compared to other encapsulation systems. The advantages of the SLN delivery system are that it can control the release of active compounds, increase the stability of active compounds, combine hydrophilic and lipophilic components, avoid toxic drug carriers, and also avoid the use of organic solvents [51,52]. 

SLNs can also be produced on a large scale in a technically or economically feasible manner [53]. SLNs are easy to prepare, require low cost, have chemical versatility, easy lipid biodegradability, are less expensive than polymer carriers, are easier to obtain approval for, and have reliability and lipid biodegradability [54,55]. This potentially beneficial effect is due to several physicochemical characteristics associated with the physicochemical properties of the lipid phase. First, the rate of degradation of the active compound can be inhibited, because its movement in the solid lipid matrix is lower than in the liquid matrix. Second, degradation reactions of active compounds are prevented by inhibiting the accumulation of active compounds on the surface of lipid particles, due to the separation and control of the active ingredients and carrier lipids in the nanoparticles. Third, there is increased absorption of some free active compounds that are difficult to absorb, after incorporation into SLNs [56,57].

The physical properties of solid lipids greatly affect the characteristics of SLNs. The use of lipids with high melting temperatures can form SLNs with large sizes due to the high viscosity of the diffuse phase. The use of solid lipids with high melting points can also increase the polydispersity index, which indicates that the particle size of SLNs is not uniform [45,58]. The length of the fatty acid chains in lipids also influences the characteristics of the SLNs. The use of solid lipids that have long chains can reduce the zeta potential, which then causes the stability of the dispersion system to decrease. A decrease in zeta potential can cause SLN particles to easily stick to one another (agglomeration occurs), which results in larger particle size [47,59]. 

However, SLNs also have several weaknesses that make it necessary to carry out additional research and development; these include the limited number of components that can be bound, the appearance of component damage or leakage during storage due to changes in the crystallinity of solid lipids, low loading capacity for some active compounds, and the need for water in sufficient quantities to dissolve SLNs [60]. SLNs can also experience lipid particle growth due to agglomeration or coalescence between dispersed SLNs, unpredictable polymer transition dynamics, the emergence of a tendency towards unpredictable gelation processes, and low binding ability resulting from the dense lipid crystal structure [7,10]. This deficiency is then exploited and developed in the next generation of nanoparticles by combining with other materials, such as liquid lipids, to improve the microstructure in the nanoparticles, which are often known as nanostructured lipid carriers (NLCs) [10,61].

### 2.2. The Differences between SLNs and NLCs

NLCs are the further development of SLNs which have drawbacks: the low diffusion rate requires a longer release time; the high water content in the system can cause crystallization, which results in a decrease in the solubility of bioactive compounds, so that they are released suddenly or burst upon release [62,63]. Formulation in the form of NLCs was carried out to increase the loading of the active components in the lipid matrix, prevent leakage in the lipid matrix, improve effectiveness in increasing the loading of active compounds, and modulate the release of active compounds. This development or modification is carried out by mixing solid lipids and liquid lipids or oils [64,65]. The incorporation of active compounds or drugs into SLNs and NLCs can be described by three models, as shown in Figure 1.

Generally, the components in the formulation of NLCs consist of solid lipids, liquid lipids, surfactants, and cosurfactants. Commonly used solid lipids include glyceryl monostearate, glyceryl behenate, cetyl alcohol, and stearic acid [64,67]. Liquid lipids that can be used include isopropyl myristate and oleic acid. Commonly used surfactants are polysorbate 80 and poloxamer 188. Commonly used cosurfactants are ethanol, propylene glycol, glycerin, and sucrose stearate [68,69]. NLCs are formulated by forming a solid lipid matrix at room temperature and at body temperature, but inside the solid lipid matrix, there are portions filled with liquid lipids. The ratio of solid lipids and oils used in the formulation of stable NLCs can be 70:30 to 99.9:0.1 [38]. The different composition of NLCs that use liquid lipids causes additional gaps in the NLC structure, which makes the loading capacity of NLCs higher than that of SLNs. However, the shape of the nanoparticles in NLCs is less compact than that in SLNs. This is caused by the inhibition of crystallization by liquid lipids in NLCs. The morphological structure of the nanoparticles in SLNs and NLCs can be identified using a scanning electron microscope (SEM) and a transmission electron microscope (TEM), as shown in Figure 2.

NLCs of active compounds are widely used in the pharmaceutical field in drug delivery systems because of their ability to control the precise release of drugs or active compounds on the target, with their nanoparticle size also causing the bioactive components to be well-encapsulated [72,73]. The development of compatible NLCs in the food sector has not been widely carried out because it is related to the food safety standards of the ingredients used. The materials for making NLCs must be food-grade and included in the GRAS (Generally Recognized as Safe) category [62,74]. The differences between SLNs and NLCs can be seen in Table 1.

NLCs have the advantage of being able to contain a greater number of drug molecules or active compounds than SLNs while minimizing damage to active compounds during storage, and can prevent particle coalescence [75]. However, the weakness of these NLCs is that there are some compounds that are not properly formulated in NLC preparations [76,77]. The high ability of NLCs to bind to active components makes NLCs suitable for use in drug delivery in high doses, so NLCs are effectively applied for supplementation compared to the fortification of food products. The release mechanism of NLCs incorporated in cream preparations can accelerate the diffusion process through the stratum corneum, which is facilitated by a matrix of NLCs that contain a lot of lipids due to their similar polarity properties [64,69]. In the preparation of the lipid matrix for NLCs, the crystal structure leads to many crystal imperfections, creating a lot of space for the placement of drug molecules and active compound molecules, whereas in SLNs, the crystal structure is more regular, so there tend to be fewer drug molecules in the lipid matrix [36,38].

## 3. Fabrication Methods for SLNs

There are various methods that can be used in the synthesis/fabrication of SLNs, including high-speed homogenization and ultrasonication, high-pressure homogenization at high or low temperatures, solvent evaporation, supercritical fluid extraction of emulsions (SFEE), multiple emulsions, and spray-drying [10,21].

### 3.1. High-Shear Homogenization/High-Speed Homogenization and Ultrasonication

High-shear homogenization/high-speed homogenization and ultrasonication is a very popular and relatively easy SLN fabrication method. The high-shear homogenization method was initially used to produce solid lipid nanodispersions, but the particle size formed after the dispersion was still micro-sized [78]. Higher stirring speed does not always significantly change the particle size but slightly increases the polydispersity index [79]. However, this method can be combined with ultrasonication by utilizing cavitation energy at high frequencies, which can help reduce dispersion particles to nano-sized. The high-shear homogenization method has several advantages; namely, it can be used for large-scale production, does not use organic solvents, and increases product stability and loading of drugs or active compounds [10,13]. The high-shear homogenization method is generally carried out at hot temperatures, so it is often also called the hot homogenization technique. The schematic representation of the ultrasonication method and high-speed homogenization can be seen in Figure 3.

The ultrasonication method and high-speed homogenization can be used in the fabrication of SLNs effectively [45,81,82]. The initial step is usually conducted by mixing the components of the active compounds that are put into the melted lipid, and then the liquid phase (which has been heated at the same temperature) is added to the previous mixture and emulsified with ultrasonication or by using a high-speed stirrer. Another way this can be conducted is by dripping the water phase on the lipid phase accompanied by high-speed stirring, and then an ultrasonicated pre-emulsion is obtained [83]. The production temperature is kept at least 5 °C above the melting point of the lipid to prevent recrystallization during the fabrication process. The nanoemulsion (o/w) obtained is filtered through a membrane to remove impurities carried during ultrasonication, and the resulting SLNs are then stored at 4 °C. Improvement of formulation stability can be conducted by the lyophilization process, which produces SLN powder by the freeze-drying process, and at a certain point, cryoprotectants such as trehalose or mannitol (5%) can be added to the SLNs as a form of protection [84].

Fabrication of SLNs with this method is relatively simple and does not require high concentrations of surfactants; the equipment used can be easily found in the laboratory [85]. This makes this method widely used for the application of SLNs as coatings and delivery systems for various drugs and active compounds. However, this method has a weakness; namely, the resulting particle size distribution is sometimes not uniform, thus affecting its stability during storage. In addition, this method also has the potential for metal contamination due to the ultrasonication process. The manufacture of SLNs with the ultrasonication method was then developed to form a stable formulation, where the combination of the ultrasonication method and high-speed homogenization at a high temperature can form smaller SLN particles by reducing the surface tension on the nanoparticles [81,86]. The ultrasonication method for SLN fabrication produces nanoparticles that are more optimal when combined with other methods, such as high-shear homogenization. Thus, the weakness of the ultrasonication method can be overcome [7,10].

The lipids used in the ultrasonication and high-speed homogenization methods include glyceryl behenate and tri-behenate (Compritol), glycerol monostearate (GMS), tristearin glyceride, and stearic acid [21]. At the same time, the surfactants that can be used are Poloxamer 188, Lecithin, and Tween [79,87]. The synthesis of SLNs using the ultrasonication method begins by dissolving the active compound to be encapsulated with melted lipid, and then the mixture is heated to 10 °C above the lipid melting point. The mixture of lipids and active compounds is then mixed into a liquid phase containing preheated surfactants and cosurfactants; the mixture is then heated to the same temperature as before and stirred slowly until a pre-emulsion is formed [52,87].

The pre-emulsion formed is then emulsified with an ultrasonicator at a stress efficiency of 35–40% for 3–5 min [87,88]. The combination of methods can be carried out before the ultrasonication process, as reported by Woo et al. [89] reported that the homogenization process can be carried out with a homogenizer on a mixture of lipids and surfactants before ultrasonication is carried out. The homogenization technique that can be conducted is high-shear homogenization or high-speed homogenization. Pre-emulsion with a high-shear homogenization treatment can be carried out at 13,000 rpm for 4 min [90], or stirred at 10,000 rpm using a high-speed homogenizer [79].

The emulsion formed after the ultrasonication process has a high temperature, so it needs to be poured into cold water (1–4 °C) and stirred using a magnetic stirrer for 3 min to crystallize the SLNs to obtain a nanoparticle suspension [79]. The crystallized SLNs are collected by centrifugation at 12,000 rpm for 60 min at 4 °C [88]. SLNs ware then lyophilized using a freeze-dryer and stored in a refrigerator [45,79]. High-shear homogenization combined with ultrasonication is the recommended method and has been widely used because of the ease of preparation and good results. Various uses of ultrasonication and high-shear homogenization methods in the fabrication of SLNs can be seen in Table 2.

### 3.2. High-Pressure Homogenization

High-pressure homogenization (HPH) is one of the popular methods among the methods that have been developed. This method can be used by industry in the manufacture of SLNs on a large scale. The pressure used in this process ranges from 100 to 2000 bar, with fat content in the formula reaching 40% [92]. There are two approaches to SLN fabrication using the HPH method, namely, hot and cold processes [37]. In the hot homogenization process, the temperature conditions are set at 5–10 °C above the melting point of the fat used. The micronutrient components or active compounds are dissolved in the melted lipid and dispersed in the aqueous surfactant phase with an Ultra-Turrax mixer, which will form a pre-emulsion. The use of heat causes the viscosity to decrease due to an increase in temperature which results in smaller and more uniform particle sizes. In this process, the quality of the pre-emulsion affects the final product of nanoparticles. Therefore, temperature and pressure must be regulated to avoid degradation of the active ingredients contained [93,94]. 

In the cold homogenization method, the micronutrient components or active compounds are dissolved in melted lipids and then rapidly cooled with dry ice or liquid nitrogen [35,95]. The cooled fat is crushed by ball milling to produce microparticles (50–100 µm) which can be dispersed in the cold surfactant phase and form pre-suspensions. The pre-suspension is homogenized in a high-pressure reactor under cold conditions to make it a dispersion system. The HPH process is continued until nanoparticles are produced. The cold homogenization technique has been extended to solve the problem of the hot homogenization technique, especially to prevent damage to the active compounds due to the use of high temperatures [8,84].

### 3.3. Solvent Evaporation

SLN fabrication by this method is carried out by means of emulsification by adding fat that has been mixed in the o/w emulsion system. The lipophilic ingredients are then dissolved in a mixture of water and a non-polar organic solvent (cyclohexane). The organic solvent is evaporated under mechanical stirring or pressure-reducing treatment to obtain fat microparticles. The fat microparticles are precipitated again to produce nanoparticles [60]. Using this method, Sjostrom [96] produced cholesterol acetate nanoparticles with sizes between 25 and 100 nm with an emulsifier, namely lecithin/sodium glycocholate. These results have been proven by Westesen and Siekmann [97], who produced phospholipid-stabilized solid lipid nanoparticles with the formation of lipid o/w emulsions with good results.

The SLNs produced by this method are greatly affected by the combination of fat with surfactants or emulsifiers as well as the concentration of fat used in the formula. This method is suitable for use on active components that have thermosensitive properties due to the production process being carried out at low temperatures, and can be used to combine hydrophilic components with o/w/o emulsions [8,98]. However, the drawback of this method is that the use of organic solvents can be toxic, so it is necessary to look for non-toxic solvents [23,62,99].

### 3.4. Other Methods

Other fabrication methods that have been developed are the supercritical fluid technique, double-emulsion method, and spray-drying technique. The use of the supercritical fluid technique in SLN fabrication has the advantage of not using solvents, so that the fabrication process can be faster and safer than other methods that use solvents [100,101]. There are several innovations for nanoparticle fabrication, including SLNs, which can be produced quickly using supercritical carbon dioxide solutions (RESS) techniques. The use of carbon dioxide (99.99%) as a substitute for solvent in this method can produce SLNs with the best results [13,84]

The preparation of SLNs by the double-emulsion method is based on the evaporation of the emulsification of the solvent that has been used to incorporate the hydrophilic compound into the SLNs [102,103]. The double emulsion w/o/w can be made with a two-step process. First, a w/o microemulsion is prepared by adding a solution containing the active component to a mixture consisting of melted lipid and surfactant/cosurfactant, at a temperature slightly above the melting point of the lipid, to obtain a homogeneous microemulsion system. The surfactants used to form w/o emulsions can be lecithin or monoglycerides. Then, the w/o microemulsion formed is added to a mixture of water, surfactant, and cosurfactant to form a w/o/w double emulsion [43,45]. Generally, Tween 80 is used to form w/o/w emulsions. The preparation of SLNs by the double-emulsion method can be seen in Figure 4. Hydrophilic compounds that are added to SLNs need to be protected or stabilized with a stabilizer to prevent partitioning into the aqueous phase during the solvent evaporation process. SLNs are then centrifuged at 12,000× *g* for 30 min at a low temperature (±4 °C) [84].

The use of the spray-dryer techniques is carried out as an alternative to the lyophilization process, in which the process changes SLNs from liquid to solid (powder) form [13,84]. Freitas and Müller [105] recommend using lipids with a melting point > 70 °C for spray-drying. The best results were obtained with a concentration of 1% SLNs in an aqueous trehalose solution or 20% trehalose in an ethanol–water mixture (10/90 *v*/*v*). The spray-drying technique is indeed cheaper when compared to lyophilization but causes particle aggregation due to high temperatures, shear strength, and partial melting of particles [106,107]. Therefore, it is necessary to use lipids with a high melting point or with the addition of trehalose [13,84].

Each method for fabricating SLNs has advantages and disadvantages, so adjustments are needed to the nature of the active ingredients used, the availability of tools and materials, and the costs involved. In general, a comparison of the advantages and disadvantages of several SLN fabrication methods can be seen in Table 3.

## 4. Applications of SLNs in Various Fields

Solid lipid nanoparticles (SLNs) have a very wide range of applications, including for topical, pulmonary, dermal, and parenteral drug delivery. These SLN application products have also been developed to minimize the toxic side effects of highly potent drugs and enhance treatment efficacy. Moreover, they have shown good potential in gene transfer, cosmetics, and food industries [10,84].

### 4.1. SLNs in Various Medical Fields and Cosmetics

The application of SLNs in the medical field is usually for oral drug administration in order to increase the bioavailability of drugs that initially have low bioavailability due to the poor water solubility of drugs [108]. In addition, the application of SLNs can also increase drug release in the body, increase residence time and lymphatic absorption, and potentially increase the bioavailability of drugs that are poorly soluble in water, especially lipophilic drugs [109,110].

SLNs can also be applied in the cosmetic and dermatological fields because of the similarity of the SLN formulation with the structure of the skin, and their application also does not show any interference and toxic effects when used topically [111,112,113,114,115]. The application of SLNs in this field is usually for sunscreen, anti-acne, and anti-aging skin care. SLNs can protect sensitive compounds against chemical degradation, increase skin moisture content, and can also penetrate active substances in the skin, inhibit UV, and hydrate the skin [10].

One of the mechanisms for releasing SLNs in the field of drug delivery occurs in the application of topical preparations, where SLNs can form an occlusive layer on the skin surface which is affected by the size of the SLNs, which are less than 400 nm [116]. When SLNs are applied to the skin surface, the water contained in the preparation will evaporate and leave an adhesive layer covering the skin, thereby reducing transepidermal water loss, which can cause the drug to penetrate into the deepest layers of the skin, reducing the corneocyte density and widening the inter-corneocyte gap [94,117].

### 4.2. SLNs in Various Food Products and as the Delivery System for Drugs or Active Compounds

Another application of SLNs is in the food sector, because they can be excellent potential carriers for sensitive compounds in the food industry to improve food quality and nutrition [118]. SLNs have also been applied for the fortification with nutrients that are easily damaged by environmental influences or nutrients that have an unfavorable taste and aftertaste that need to be protected or encapsulated in the form of SLNs [45]. In addition to carrying nutrients, SLNs have also been widely used as a drug delivery agent or as a carrier for other active compounds that are beneficial to health, such as medicinal compounds and antioxidants [119,120,121]. SLNs have been widely applied for delivery of drugs for neurological diseases and cancer, and for preventing antibiotic resistance, because this carrier can contain many antimicrobial drugs, increase drug absorption, and reduce bacterial efflux. SLNs are a good alternative in terms of functionality and biocompatibility, because they can directly reach the site of infection, increase drug bioavailability, and reduce toxicity or side effects [122].

SLNs are lipid-based drug delivery systems that have distinctive properties, such as large surface area, high drug loading, and protection of the drug or active compounds from the environment [84,98,123,124]. In addition, SLNs can increase the absorption efficiency or bioavailability of drug delivery systems. SLNs can have two advantages in drug delivery systems, namely in the effective protection of drugs and active compounds during the formulation process due to increased encapsulation efficiency; the effect of this system is preferred because of the increased release of drugs and active compounds due to the homogenization process in the production of SLNs [51,125,126].

The use of SLNs as an active compound delivery system is included in the solid fat material model, and the enriched nuclei are the desired active compounds. In drug delivery systems, molecular dispersion occurs in a solid lipid matrix when the particles are formed by surfactants or cosurfactants using cold homogenization techniques. Drug delivery with SLNs has a very strong interaction, especially in its lipid components [124,127]. In a drug delivery system with SLNs, the solid lipid core is formed after the lipid recrystallization temperature is reached, and the concentration of the SLN output will melt due to dispersion when the temperature decreases [84,128]. The essence of the drug delivery system with SLNs is the cooling of the nano- or microemulsion, which results in supersaturation of the drug delivery dissolved by the melting lipid approaching its saturation solubility, and precipitation will occur before lipid recrystallization. The extra cooling eventually leads to recrystallization of the lipid that surrounds the drug delivery system as a thin, membrane-like layer [8,129].

The development of the SLN system for various active compounds aims mainly to increase the loading capacity, solubility, and stability [8,130]. Several studies have been carried out on the utilization or application of SLNs related to the absorption and release of the active compound core ingredients in them. Pople and Singh [131] reported that SLNs that contains vitamin A have better release penetration than conventional preparations; namely, the concentration is doubled compared to conventional gels and also has good stability in its application to drug use. Another study, conducted by Kim et al. [132], showed that cyclosporin manufactured in the form of SLNs had good penetration and increased release two times higher than conventional preparations in vitro. 

Lipid nanoparticles, including SLNs, have also been widely applied in mRNA delivery, especially used in mRNA vaccines, which have been proven safe and effective after going through clinical trials [133,134]. This development has also been applied to the provision of the vaccine against COVID-19, which was once a pandemic in various countries. Various lipid nanoparticle formulas for mRNA vaccines were developed, starting from the constituent lipid matrix, the surfactant used, and the encapsulated mRNA or active ingredient. Francis et al. [135] applied SLNs to non-viral DNA vaccine delivery to DNA vaccine candidates encoding the Urease alpha (UreA) using Monophosphoryl lipid A. It was found that SLNs have particle uptake by cells within 3 h in the endosome compartment. SLNs were able to stimulate the expression of the pro-inflammatory cytokine TNF-α in THP-1 cells. Lou et al. [136], encapsulating mRNA vaccine encoding the rabies virus glycoprotein (RVG) in the form of lipid nanoparticles using cationic lipids, found that lipid nanoparticles elicited strong humoral and cellular immune responses at a dose of 1.5 µg and have the potential to become and efficient form of mRNA vaccine delivery. The applications of solid lipid nanoparticles in various fields can be seen in Table 4. 

## 5. Solid Lipid Nanoparticles for the Encapsulation of Antioxidant Compounds

SLNs have been widely applied to encapsulate various active compounds and antioxidants, both polar (hydrophilic), semipolar, and non-polar (hydrophobic) antioxidants [11,121,147]. Fabrication of SLNs of polar and semipolar antioxidant compounds requires surfactants or emulsifiers to bridge between lipid encapsulants and polar core materials. Meanwhile, non-polar antioxidant compounds tend to be more easily encapsulated in SLNs, along with the compatibility of the hydrophobicity properties between the lipid matrix and the core compound [128,148]. Various applications of SLNs for the encapsulation of non-polar antioxidant compounds can be seen in Table 5.

Various types of antioxidant compounds have been successfully encapsulated in the form of SLNs with various types of lipid matrices and fabrication methods. Jain et al. [120] have succeeded in making SLN β-Carotene with a lipid matrix in the form of glyceryl monostearate and phospholipid S-100 emulsifier, using the hot homogenization process by the high-shear mixer method. The SLNs obtained were able to increase the stability of the antioxidant β-Carotene 3 times higher than without encapsulation, and SLNs also increased the bioavailability and efficacy of β-Carotene. Schjoerring-Thyssen et al. [149] also succeeded in synthesizing SLNs of β-Carotene, using the hot-melt high-pressure homogenization method, with fully hydrogenated sunflower oil as a lipid matrix. The SLNs of β-Carotene obtained have stable crystals in the form of β-crystals.

SLNs are also effective for encapsulating the active compound curcumin, which is an antioxidant [152,161]. Curcumin can be encapsulated in the form of SLNs using the micro-emulsification technique method with Compritol 888 as a lipid matrix and assisted by emulsifiers in the form of Tween 80, lecithin, and polysorbate 80. Curcumin-SLNs increased oral bioavailability and stability up to 155 times more than free curcumin [150]. Shandu et al. [151] also succeeded in encapsulating curcumin in the form of SLNs using the hot-high-pressure homogenization method with a lipid matrix in the form of Compritol^®^ 888 ATO and an emulsifier in the form of soya lecithin. Curcumin-SLNs have an entrapment efficiency of 75% and have a significant antimicrobial effect.

SLNs are also effective for encapsulating tocopherols as a source of antioxidants. Shylaja and Mathew [153] succeeded in synthesizing SLN-alpha tocopherol, using the method of hot homogenization with a high-shear homogenizer and stearic acid as the lipid matrix. SLN tocopherol has an in vitro release of 74.33%, entrapment efficiency of up to 98.67%, and is stable during storage. SLN-alpha tocopherol can also be synthesized using the hot high homogenization with the high-pressure homogenizer method with a lipid matrix in the form of glyceryl behenate, which produces SLNs with a high α-tocopherol recovery rate, stable during storage [154]. At the same time, Oehlke et al. [121] succeeded in encapsulating alpha-tocopherol and ferulic acid in the form of SLNs with a lipid matrix in the form of glyceryl tristearate, using the ultrasound-assisted hot emulsification method. The SLNs obtained were stable for 15 weeks, and the significant radical scavenging activity was maintained.

Other antioxidant compounds were also successfully encapsulated in the form of SLNs, including sesamol, which was encapsulated using the microemulsion technique with a lipid matrix in the form of glyceryl monostearate and an emulsifier in the form of α-phosphatidylcholine. Sesamol-SLNs applied to cream manufacture showed retention in the skin with minimal flux and showed normalization of post-induced skin cancer [155]. Meanwhile, Lacatusu et al. [157] encapsulated umbelliferone using n-Hexadecyl palmitate and glyceryl stearate with the high-shear homogenization method; it was found that the SLNs had a good entrapment efficiency and antioxidant properties up to 75%. SLN technology still has many opportunities to encapsulate various other non-polar active compounds, because the affinity and hydrophobicity properties are very suitable for the solid lipid matrix used. However, SLN fabrication for the encapsulation of non-polar antioxidant compounds needs to pay attention to several factors that can have an effect. The selected fabrication method should avoid using too high temperatures to keep the antioxidant compounds from being damaged. The lipid used should have high lipophilicity. The more lipophilic a solid lipid is, the more it can accommodate more lipophilic active ingredients [162]. If using a surfactant, one must be chosen that has a low HLB in order to homogenize the system, which is dominated by hydrophobic compounds [17]. 

SLNs have also been widely applied to encapsulate various polar (hydrophilic) and semipolar antioxidant compounds [163,164,165,166]. Antioxidant compounds that are polar tend to be more difficult to encapsulate in SLNs due to differences in polarity between the lipid matrix and the core compounds, thus requiring a suitable emulsifier and appropriate fabrication techniques or methods for high-efficiency entrapment, small particle size, and homogeneous or uniform [11,167]. Most of the active compounds that are antioxidants are polar or semipolar in nature, such as from a group of phenolic compounds, flavonoids, water-soluble vitamins, and several essential minerals. Therefore, researchers are trying to continue to develop various applications of SLNs for the encapsulation of polar and semipolar antioxidant compounds, which can be seen in Table 6.

Phenolic, polyphenolic, and flavonoid compounds are polar-semipolar active compounds that are abundant in various plants and have high antioxidant activity [181,182]. Many efforts have been made to protect and stabilize the antioxidant activity of phenolic and polyphenolic compounds, including through encapsulation in the form of SLNs. Epigallocatechin-3-gallate (EGCG) has been successfully encapsulated in the form of SLNs by various methods and lipid matrices. Shtay et al. [168] synthesized EGCG-SLNs, using the hot homogenization method and cocoa butter as a lipid matrix; they obtained EGCG-SLNs with an entrapment efficiency of 68.5% and increased the shelf life. Ramesh and Mandal [163] synthesized EGCG SLNs, using the same method but with a lipid matrix glycerol monostearate and an emulsifier in the form of poloxamer 188; SLNs were obtained with the entrapment efficiency of 81% and were able to increase the bioavailability and stability of EGCG. Meanwhile, Radhakrishnan et al. [169] synthesized EGCG-SLNs, using the emulsion-solvent evaporation method with lipid matrices in the form of stearic acid, tripalmitin, and tristearin; the obtained EGCG-SLNs had cytotoxicity of 8.1-fold higher against MDA-MB 231 human breast cancer cells compared to free EGCG. The myricitrin polyphenol compound has also been successfully fabricated in the form of SLNs, which were able to improve its bioactivity. Ahangarpour et al. [170] reported that myricitrin was encapsulated using lipid matrices in the form of Compritol^®^ 888 and oleic acid with the cold homogenization method, showing that SLN-myricitrin had antioxidant, antidiabetic, and antiapoptotic effects in vivo and in vitro. SLN-myricitrin applied to type-2 diabetic male mice showed that SLN-myricitrin reduced oxidative stress and increased antioxidant enzyme levels [171].

Phenolic compounds that have high antioxidant activity include resveratrol, gallic acid, and quercetin. These compounds need to be protected for their bioactivity, including through encapsulation in the form of SLNs. Gokce et al. [172] succeeded in fabricating resveratrol-SLNs with glyceryl behenate as a lipid matrix, using high-shear homogenization. Resveratrol-SLNs were effective in reducing ROS accumulation and exerting antioxidant activity, and the drug in the shell model was relevant. Another study by Neves et al. [173] reported that resveratrol-SLNs could be fabricated with cetyl palmitate as a lipid matrix and polysorbate 60 as an emulsifier, using a combination of high-shear homogenization and the ultrasound method. Resveratrol-SLNs increased oral bioavailability, stability, and sustained release. Gallic acid phenolic compounds were also successfully fabricated in the form of SLNs by Subroto et al. [164]; they used stearic acid and fat rich in monolaurin, using the double-emulsion method assisted by high-speed homogenization and ultrasonication. In this fabrication, fat-rich monolaurin acts as a solid lipid that is healthy as well as an emulsifier due to the same properties as other monoacylglycerols, which act as surfactants [183,184,185]. Gallic acid-SLNs had an entrapment efficiency of 93.75% and were able to increase the antioxidant activity of a chocolate bar. Meanwhile, the phenolic quercetin compound was successfully fabricated in the form of SLNs by Li et al. [174] with glyceryl monostearate, using the method of emulsification and low-temperature solidification. SLN-Quercetin had a particle size of about 155.3 nm, with a spherical shape, entrapment efficiency of 91.1%, and relative bioavailability of 571.4%.

One of the antioxidant compounds that have high activity and can be produced by the human body is ubiquinone or coenzyme Q10. Coenzyme Q10 is semipolar, and the compound is also effective and stable when encapsulated in the form of SLNs. Farboud et al. [175] fabricated Coenzyme Q10-SLNs with cetyl palmitate and stearic acid, using the high-speed and high-pressure homogenization method to obtain SLNs with particle sizes 50–100 nm; and polydispersity index was 0.11–0.20. At the same time, Gokce et al. [176] reported that Coenzyme Q10-SLNs, which were fabricated with glyceryl behenate using high-shear homogenization, were able to enhance cell proliferation more than free Coenzyme Q10, but performed no better than liposomes.

Water-soluble high-activity antioxidant compounds that are easy to find are anthocyanins and vitamin C or ascorbic acid. These compounds can also be encapsulated in the form of SLNs. Ravanfar et al. [179] reported that anthocyanin-SLNs could be fabricated with palmitic acid, pluronic F127, egg lecithin, and span 85, using the microemulsion dilution method to obtain anthocyanin-SLNs that had entrapment efficiency of 89.2 and effectively increased stability. Meanwhile, ascorbic acid-SLNs were successfully fabricated by Güney et al. [166] with compritol^®^ and Tween 80, using the hot homogenization method. Ascorbic acid-SLNs showed high entrapment efficiency, sustained release, high cytotoxic activity against H-Ras 5RP7 cells compared to free AA, more efficient cellular uptake, and induced apoptosis.

The success of SLN fabrication for polar antioxidant compounds is determined by the method chosen. The method used should not involve high temperatures to avoid thermal damage. The preferred method is primarily emulsion formation, such as double emulsion (W1/O/W2). Emulsion formation will help SLN homogeneity and increase entrapment efficiency [45,186]. The selection of the right type of surfactant also determines the success of SLN fabrication. The more polar the encapsulated antioxidant compound, the more surfactant or emulsifier with a higher HLB is needed, so that it can improve the homogeneity of the emulsion system, lead to a smaller particle size, and increase absorption efficiency [187]. 

## 6. Conclusions and Future Research

Solid lipid nanoparticles can be used as an encapsulation system and delivery system for various active compounds that tend to be lipophilic, but currently, it has been widely applied to hydrophilic active compounds by various methods, especially by forming a double emulsion w/o/w or by using an emulsifier. The success of SLN fabrication is largely determined by several factors, such as the choice of the fabrication method, the type of lipid matrices used, and the selection of surfactants or emulsifiers, as well as the preparation conditions and processes used. SLN fabrication has been effective in coating various active compounds. SLNs have been applied to various active compounds such as drugs, minerals, and various polar, semipolar, and non-polar antioxidant compounds, but several combinations of materials and processes need to be carried out to produce more stable SLNs and high entrapment efficiency.

Antioxidant compounds have attracted the attention of researchers, along with the need for antioxidants that are stable and have high bioavailability to prevent and treat various degenerative diseases. Various antioxidant compounds are numerous and have the potential to continue to be developed for fabrication in the form of SLNs using various types of lipid matrices, emulsifiers, and various methods or combinations thereof. These antioxidant-SLNs play a very large role in various fields of food, pharmaceuticals, and cosmetics to provide high-activity, stable antioxidant compounds, and to become the right delivery system for various treatments and fulfillment of appropriate nutrition. Aspects that are still a challenge in the future include efforts to make fabrication economically efficient, studies on increasing biocompatibility with other components, achieving active compounds on target, and clinical trials before they are used by the public.

## Figures and Tables

**Figure 1 antioxidants-12-00633-f001:**
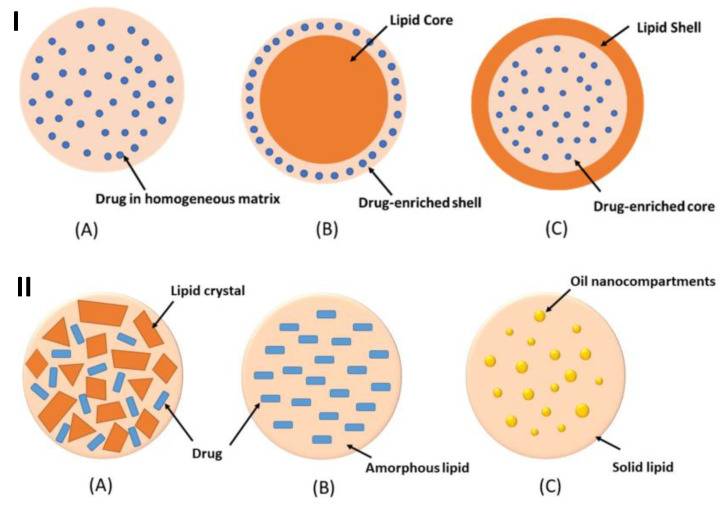
The active compounds or drug incorporation models of SLNs (**I**): homogeneous matrix (**A**), drug-enriched shell (**B**), and drug-enriched core (**C**); and NLCs (**II**): imperfect crystal type (**A**), amorphous type (**B**), and multiple-oil-in-fat-in-water type (**C**) [66].

**Figure 2 antioxidants-12-00633-f002:**
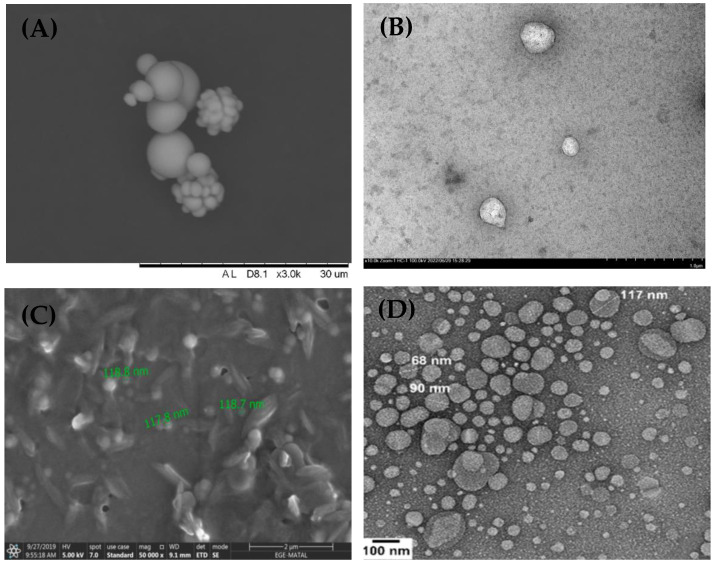
The morphological characterization of SLNs by SEM (**A**) and TEM (**B**) [45]. The morphological characterization of NLCs by SEM (**C**) [70], and TEM (**D**) [71].

**Figure 3 antioxidants-12-00633-f003:**
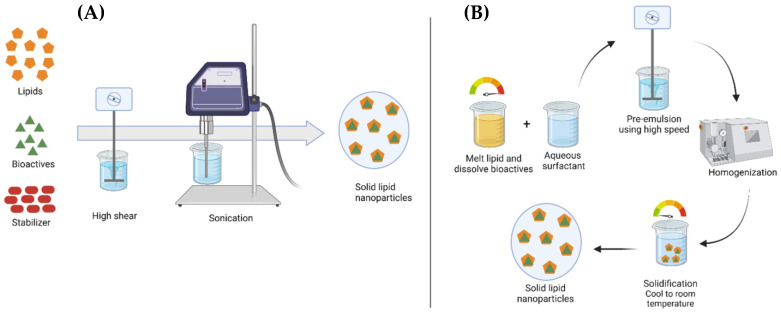
Schematic representation of ultrasonication method (**A**) and hot homogenization technique (**B**) [80].

**Figure 4 antioxidants-12-00633-f004:**
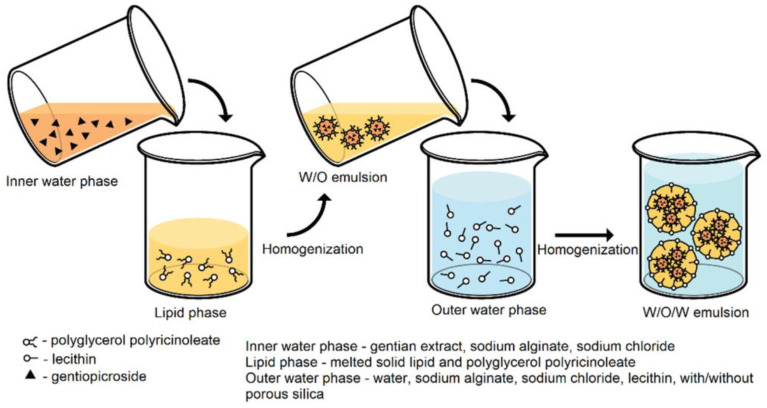
The preparation of SLNs by the double-emulsion method [104].

**Table 1 antioxidants-12-00633-t001:** The differences between SLNs and NLCs.

Characteristics	SLNs	NLCs
Ingredients	Solid lipids, surfactants, cosurfactants (optional), and active compounds	Solid lipids, liquid lipids, surfactants, cosurfactants (optional), and active compounds
Active compounds	Hydrophilic and hydrophobic compounds	Suitable for hydrophobic compounds
Crystal structure	Regular and perfect	Irregular, imperfection, lots of free space
Lipid matrix ability to bind to active compounds	Little/slight	More than SLNs
Application	Fortification	Supplementation
Cost	Inexpensive	Expensive
Fabrication difficulty	Easy	Difficult

**Table 2 antioxidants-12-00633-t002:** Various uses of ultrasonication and high-shear homogenization methods in the fabrication of SLNs of various active compounds.

Method/Treatment	Materials	Product Characteristics	References
Ultrasonication	Lipids: stearic acid; Surfactant: Tween 80; Cosurfactant: NaDC; Active substance: Ciprofloxacin	The particle size of SLNs is about 163–369 nm, EE of 51.25–90.08%, depending on lipid composition and surfactant.	[87]
Ultrasonication	Lipids: stearic acid;Active substance: Enrofloxacin	The SLN formed has an entrapment efficiency (EE) of 70.56% with diameter of 217.3 nm.	[88]
Hot homogenization and ultrasonication	Lipid: stearic acid; Surfactant: Tween 60; Active substance: salicylic acid	The particle size of SLNs is about 194–255 nm, EE values are in the range of 49–69%	[89]
Hot homogenization and ultrasonication	Lipid: stearic acid and fat rich in monolaurin; Surfactant: Tween 80; Active substance: Ferrous sulfate	The Z-average of SLNs ranges from 278.7 to 540.4 nm, polydispersity index of 0.88–1.24, and EE values are in the range of 99.7–99.9%	[45]
High-shear homogenization and ultrasonication	Lipids: stearic acid; Surfactant: Quillaja saponin; and Active substance: Imatinib mesylate,	The particle size of SLNs ranges from 143.5 to 641.9 nm, EE of 41–66.2%, polydispersity index (PI) of 0.127–0.237, and Zeta potential (ZP) of −2.43–0.95 mV.	[90]
High-shear homogenization and ultrasonication	Lipids: stearic acid; Surfactant: Tween80; Active substance: Voriconazole	The particle size of SLNs ranges from 286.6 to 313.1 nm, and ZP of −15 mV to −11 mV	[91]
High speed homogenization and ultrasonication	Lipid: stearic acid, Surfactant: Poloxamer 188, and Active substance: candesartan cilexetil (CDC)	The optimal SLN particle size is 197.9 nm, and ZP is about −21.3 mV	[79]
High speed homogenization and ultrasonication	Lipids: stearic acid, glycerol monostearate and Apifil^®^, Surfactants: Tego care and Planta care; active ingredients: Ceramide	Stable SLN ceramide has a loading capacity of 4% with Apifil^®^ capacity of 4%, and Planta care 1% had a particle diameter of 113.5 nm and a polydispersity index (PI) of 0.263, and was stable for 2 months of storage with a ceramide content of 92.26%.	[52]

**Table 3 antioxidants-12-00633-t003:** Comparison of the advantages and disadvantages of several SLN fabrication methods.

Method	Advantages	Disadvantages	References
High-Pressure Homogenization	Low cost, can be produced on a small scale (laboratory) or large scale	It can cause damage to biomolecules, and on a large scale, it requires enormous energy and pressure.	[43,79]
Ultrasonication/High-Speed Homogenization	Effectively reduces surface tension, the process is easy and sustainable, can be commercialized	There is a potential for metal contamination due to the high rotational speed of the tool, and large-scale production requires a high investment.	[7,43]
High-shear homogenization	Can be done without the use of solvents, and the process is easy	The particle size is relatively large and less uniform.	[10,79]
Solvent Evaporation Method	The resulting SLNs are stable and uniform, and commercialization can be carried out.	Requires a lot of energy, biomolecule damage can occur, and there is a potential for toxic solvents.	[46,60]
Supercritical Fluid Technique	Does not require a solvent; the particles formed are powdery.	Cost is high.	[43,84]
Double-Emulsion Method	Capable of incorporating hydrophilic components	Large-sized particles at the end of the fabrication	[43,46]
Spray-Drying Technique	Lower cost than lyophilization	Possibility of particle aggregation due to high temperatures	[46,84]

**Table 4 antioxidants-12-00633-t004:** Applications of SLNs in various fields.

Field or Product Type	Fabrication Conditions	Product Characteristics	References
Skincare for hyperpigmentation	- Active ingredient: N-acetyl-d-glucosamine (NAG) - Method: high-shear homogenization - Lipid phase: hydrogenated castor oil, Peg-25, phosphatidylcholine, and cetyl palmitate - Water phase: purified water	- NAG can act as an emulsion stabilizer - Differences in concentration can affect the process of permeation and release of NAG into the skin, which can contribute to skin repair	[137]
Skincare	- Active ingredients: Epigallocatechin Gallate (EGCG), vitamin E, and Resveratrol. - Method: high-pressure homogenization - Solid lipids: compritol, phospholipon 80, and cetyl palmitate - Liquid lipids: sesame oil - Surfactant: Tween 80	Resveratrol and Vitamin E show results by providing protection against UV-induced parts to prevent degradation, but EGCG does not show the same results.	[138]
Photodynamic therapy	- Active ingredient: DH-I-180-3 - Method: solvent evaporation and hot homogenization - Lipids: Stearic acid and Capmul1 MCM C8 - Surfactant: poloxamer 188 - Cosurfactant: lecithin	The incorporation of DH-I-180-3 into SLNs enhances targeting efficacy and enhances photocytotoxicity.	[139]
Drug delivery	- Active ingredient: Oxyresveratrol (OXY) - Method: high-shear homogenization - Lipid phase: C888 - Water phase: Tween 80 and soy lecithin	Increases the bioavailability of OXY up to 125% compared to OXY that is not coated by SLN.	[140]
Drug delivery	- Active ingredient: hydrochlorothiazide (HCT) - Method: High-shear homogenization and ultrasonication - Surfactant: Precirol^®^ATO5, with two different surfactants, namely Tween 80 and Pluronic F68	- Use of Tween 80 elicits complete drug release at 150 min, without providing sustained release. - The use of pluronic F68 ensures a sustained release of more than 75% at the end of the test and a 1.8-fold increase over simple drug suspensions.	[119]
Drug delivery	- Active ingredient: Clarithromycin - Method: high-speed homogenization. - Lipid phase: stearic acid, tripalmitin, and glyceryl behenate - Surfactant: Tween 80	The best particle size was 318.0 nm with a polydispersity index of 0.228–0.472, and the Clarithromycin-SLNs showed extended release up to 48 h.	[141]
Drug delivery	- Active ingredient: Gliclazide - Method: high-shear homogenization. - Lipid phase: stearic acid - Surfactant: PEG 400 and Tween 80	Spherical-shaped particles with the best size of 745.8 nm, Polydispersity index of 0.776, and absorption efficiency of 75.29%.	[42]
Iron delivery	- Active ingredient: ferrous sulphate - Method: double emulsion and solvent evaporation. - Lipid phase: Stearic acid - Surfactant: Polyvinyl alcohol	The particle size of 300.3–495.1 nm, iron absorption indicates Caco-2 iron absorption up to 642.77 ng/mg cell protein or 24.9% greater than free ferrous sulphate, and ferrous sulphate-SLNs have the potential for iron delivery.	[142]
Encapsulation	- Active ingredients: Ferulic acid and tocopherol - Method: hot homogenization - Lipid phase: Glyceryl tristearate. - Surfactant: polysorbate 20.	- FA-SLN activity was stable during the entire storage period, and TOC-SLN activity increased with time - SLNs showed good stability and high antioxidant activity during 15 weeks of storage.	[121]
Encapsulation	- Active ingredients: ferrous sulfate - Method: ultrasonication and double emulsion - Lipid phase: glycerol - monostearate and stearic acid - Surfactant: Tween 80 and span 80	The particles are spherical in shape with a size of 358 nm, a polydispersity index of 0.154, and an entrapment efficiency of 92.3%.	[143]
Fortification	- Active ingredients: gallic acid - Method: high-shear homogenization, ultrasonication, and double emulsion - Lipid phase: stearic acid and fat rich in monoacylglycerols and diacylglycerols - Surfactant: Tween 80	Gallic acid-SLNs, which was fortified with 5% in chocolate bars, increased the antioxidant activity and total phenolic content with an IC_50_ of 174.24 and had good organoleptic characteristics.	[45]
Skincare for cellulite	- Active ingredient: Caffeine - Method: hot homogenization - Lipid phase: Precirol	SLN containing caffeine showed good stability for 12 months of storage.	[144]
Skin allergy medication	- Active ingredient: Loratadine (LRT) - Method: High-pressure homogenization - Lipid phase: Precirol - Surfactants: TegoCare 450 and 1-hexadecanol	- Produces a particle size of 11–14 nm; the highest release rate was 0.821 mcg/mL/h - Stable on storage for 6 months	[145]
DNA Vaccine Delivery	- Active ingredient: DNA vaccine candidate encoding the Urease alpha (UreA) - Method: solvent-emulsification - Lipid phase: cholesteryl oleate, glyceryl trioleate, DOPE, DC-cholesterol, and monophosphoryl lipid A.	SLNs have particle uptake by cells within 3 h in the endosome compartment. SLNs were able to stimulate the expression of the pro-inflammatory cytokine TNF-α in THP-1 cells. Lipoplexes are compatible with being transfused efficiently in murine immune cells.	[135]
mRNA Vaccine Delivery	- Active ingredient: mRNA encoding prM-E from Zika, and mRNA encoding influenza HA genes - Method: solvent-emulsification - Lipid phase: ionizable lipid MC3, cholesterol, DSPC, and PEG lipid	Lipid nanoparticles elicit a strong immune response, and the tolerability of mRNA vaccines can be increased without affecting their potency.	[146]
mRNA Vaccine Delivery	- Active ingredient: mRNA vaccine encoding the rabies virus glycoprotein (RVG) - Method: solvent-emulsification - Lipid phase: 3ß-[N-(N′,N′-dimethylaminoethane)-carbamoyl]cholesterol (DC-Chol), DDA, DOTAP, DMTAP, DSTAP, and DOBAQ	Lipid nanoparticles elicited strong humoral and cellular immune responses at a dose of 1.5 µg. Lipid nanoparticles have the potential to become an efficient method for mRNA vaccine delivery.	[136]

**Table 5 antioxidants-12-00633-t005:** Application of SLNs for the Encapsulation of Non-Polar Antioxidant Compounds.

Antioxidant Compound	Lipid Matrices	Fabrication Method	Characteristics of SLNs	References
β-Carotene	glyceryl monostearate, gelucire50/13, and phospholipid S-100	hot homogenization process with high-shear mixer	SLN increased the stability of antioxidant activity more than 3 times at 90 days. The bioavailability and efficacy also improved.	[120]
β-Carotene	fully hydrogenated sunflower oil	hot-melt high-pressure homogenization	β-Carotene formed 10 cis isomers, but did not affect the crystals of SLNs in the form of stable β-crystals.	[149]
Curcumin	Compritol 888, Tween 80, lecithin, Polysorbate 80	Micro-emulsification technique	Curcumin-SLN increased oral bioavailability and stability 32–155 times more than free curcumin, and sustained release.	[150]
Curcumin	Compritol^®^ 888 ATO, soya lecithin	Hot-high-pressure homogenization	Curcumin-SLN had an entrapment efficiency of 75%, polydispersity index of 0.143, particle size of <200 nm, and a significant antimicrobial effect.	[151]
Curcumin	glycerol behenate (Compritol^®^ 888 ATO)	Solvent evaporation method	Curcumin-pSLN has a high antioxidant activity of 3.73 ORAC Units compared to free curcumin of 1.6 ORAC Units, has a zeta potential of −30 mV, a particle size of less than 200 nm, and is stable during storage.	[152]
Alpha-tocopherol	Stearic acid	Hot homogenization by a high-shear homogenizer	SLN-alpha tocopherol had an in vitro release of 74.33% after 8 h. SLN was spherical in size <1000 nm and EE up to 98.67%.	[153]
Alpha-tocopherol	Glyceryl behenate (Compritol^®^ 888)	Hot high homogenization with a high-pressure homogenizer	SLN-α-tocopherol has a particle size of 214.5 nm, zeta potential of −41.9 mV, α-tocopherol recovery rate of 75.4%, and was stable for 21 days at 6 °C, with polymorphic forms were α and β′.	[154]
Alpha-tocopherol and Ferulic acid	Glyceryl tristearate	ultrasound-assisted hot emulsification	SLN-Tocopherol and SLN-Ferulic acid with a lading of 2.5 mg/g were stable for 15 weeks, and significant radical scavenging activity was maintained.	[121]
Sesamol	Glyceryl monostearate, Sodium deoxycholate, and α-phosphatidylcholine	Microemulsion technique	Sesamol-SLN had an entrapment efficiency of 88.21% and a particle size of 127.9 nm. Application of the cream showed retention in the skin at minimum flux, and showed normalization of post-induced skin cancer.	[155]
Sesamol	Compritol 888, Soy Lecithin, and Tween 80	Microemulsion technique	Sesamol-SLN had a particle size of about 120.3 nm, and better hepatoprotection compared to free sesamol. Sesamol-SLN improved bioavailability, reduced toxicity and irritation, and controlled the effect of entrapped sesamol.	[156]
Umbelliferone (7-hydroxycoumarin)	Glyceryl Stearate and n-Hexadecyl Palmitate	high-shear homogenization	SLN had a particle size of about 173.4 nm, good entrapment efficiency (60.70%), and antioxidant properties of 75%.	[157]
Vitamin A	Cetyl alcohol, Tween 80, and Gelucire 44/14^®^	hot homogenization	Vitamin A-SLNs have an entrapment efficiency of >90%, a particle size of about 40 nm, an average size of 30–50 nm in the spherical form, and vitamin A-SLNs have stability >2 times compared to free vitamin A.	[158]
Astaxanthin	lipid phase: stearic acid, surfactant: Poloxamer	solvent-diffusion method	Astaxanthin-SLNs have a particle size of <200 nm, were able to maintain the stability of astaxanthin during 6 months of storage, and were able to improve antioxidant capacity more than free astaxanthin.	[159]
Ascorbyl Palmitate (AP)	Lipid phase: Glyceryl monostearate (GMS), Surfactant: Pluronic-F68	Ultrasonic method	SLN-AP had a cytotoxic effect at lower concentrations than in the form of AA or DHA. Formulation of SLN-AP with 3% Pluronic F-68 produced SLNs with good physicochemical characteristics and stable antioxidants but need further optimization because of their intrinsic cytotoxicity.	[160]

**Table 6 antioxidants-12-00633-t006:** Application of SLNs for the Encapsulation of Polar and Semipolar Antioxidant Compounds.

Antioxidant Compound	Lipid Matrices	Fabrication Method	Characteristics of SLNs	References
Epigallocatechin-3-gallate (EGCG)	Cocoa butter	Hot homogenization method	EGCG-SLNs had an entrapment efficiency of about 68.5%, particle size between 108 and 122 nm, and increased shelf life.	[168]
Epigallocatechin-3-gallate (EGCG)	Glycerol monostearate, poloxamer 188, polyoxyethylene stearate	Hot melt homogenization	SLN-EGCG had a particle size of about 300.2 nm, an entrapment efficiency of about 81%, and was able to improve the bioavailability and stability of EGCG.	[163]
Epigallocatechin-3-gallate (EGCG)	Stearic acid, Tripalmitin, and tristearin	Emulsion-solvent evaporation method	EGCG-SLNs had cytotoxicity of 3.8 folds higher against human prostate cancer cells of DU-145 and 8.1 folds higher against human breast cancer cells of MDA-MB 231 compared to free EGCG.	[169]
Myricitrin	Compritol^®^888 and oleic acid	Cold homogenization	SLN-myricitrin had antioxidant, antidiabetic, and antiapoptotic effects in vivo and in vitro.	[170]
Myricitrin	Compritol^®^888 and oleic acid	Cold homogenization	SLN-myricitrin reduced oxidative stress and increased antioxidant enzymes level	[171]
Resveratrol	Glyceryl behenate (Compritol^®^ 888)	High-shear homogenization	Resveratrol-SLNs had an antioxidant activity that can reduce ROS effectively. Resveratrol-SLNs had a particle size of about 287.2 nm, ZP was −15.3 mV, and the drug was trapped in the SLNs efficiently.	[172]
Resveratrol	Cetyl palmitate, polysorbate 60	High-shear homogenization and the ultrasound method	Resveratrol-SLNs increased oral bioavailability, stability, and sustained release.	[173]
Gallic acid	Stearic acid and fat rich in monolaurin	Double emulsion assisted by high-speed homogenization and ultrasonication	Gallic acid-SLNs had particle size of 224.4–3596.3 nm, entrapment efficiency of about 93.75%, polydispersity index of about 0.85, and were able to increase the antioxidant activity of chocolate bar.	[164]
Quercetin	Glyceryl monostearate, Tween-80, Soya Lecithin	Emulsification and low-temperature solidification	SLN-Quercetin had a spherical shape, with zeta potential of −32.2 mV, drug loading of 13,2%, entrapment efficiency of about 91.1%, a particle size of 155.3 nm, and relative bioavailability of 571.4%.	[174]
Phenolic and flavonoids of *Prunus persica* (L.) ethanolic extract (PPEE)	Glyceryl monostearate, Tween 80	Solvent evaporation method	PPEE-SLNs increased the antioxidant activity, anti-tyrosinase, anti-collagenase, and anti-elastase more than free PPEE.	[165]
Coenzyme Q-10	Cetyl palmitate or stearic acid	High-speed & high-pressure homogenization	Particle sizes were 100 nm and 50 nm, with polydispersity index between 0.20 and 0.11	[175]
Coenzyme Q-10	Glyceryl behenate (Compritol^®^ 888)	High-shear homogenization	SLN-Coenzyme Q10 was able to enhance cell proliferation more than free Coenzyme Q10, but no better than liposomes; less effective against ROS accumulation.	[176]
Ofloxacin	Palmitic acid	Hot homogenization and ultrasonication	SLNs had an encapsulation efficiency of 41.36%, loading capacity of 4.40%, diameter of 156.33 nm, zeta potential of about −22.70 mv, and polydispersity index of 0.26. SLNs increased the residence time up to 43.44 h and improved the bioavailability 2.27-fold.	[177]
Tetrandrine	Glyceryl behenate, Stearyl amine, and PEG stearate	Emulsion evaporation-solidification at low temperature	Tetrandrine-loaded SLNs retained the drug entity better than free tetrandrine, had an average diameter of 18.77 nm, a zeta potential of −8.71, an AUC value 2-fold higher than free tetrandrine, and had no significant toxicity.	[178]
Anthocyanins	Palmitic acid, Pluronic F127, Span 85, and egg lecithin	Microemulsion-dilution method	Anthocyanin-SLNs had an entrapment efficiency of about 89.2, a particle size of 455 nm, and revealed spherical morphology.	[179]
Ascorbic acid (AA)	Compritol^®^ and Tween 80	hot homogenization method	AA-SLNs showed sustained release, high cytotoxic activity against H-Ras 5RP7 cells, high entrapment efficiency, more efficient cellular uptake, and induced apoptosis.	[166]
Grape seed extract (GSE) rich in Proanthocyanidins	Gelucire^®^ 50/13 and Tween 85	melt-emulsification method	GSE-SLNs had good loading efficiency of about 0.058 mg/mg, a particle size of 243, stable antioxidant activity, and favorable properties for lung delivery.	[180]

## Data Availability

Not applicable.

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
