# Peer review of "Solid Lipid Nanoparticles: Review of the Current Research on Encapsulation and Delivery Systems for Active and Antioxidant Compounds"

_antioxidants, 2023, doi:10.3390/antiox12030633_

Round 1
Reviewer 1 Report
This is a review which might very useful for someone entering the field of lipid nanoparticles. Particularly well covered is the subfield of encapsulation of antioxidant compounds.
Some minor modifications are needed in order to improve the overall impression:
1) There are no figures! I strongly recommend placing few illustrations to enhance the review readability;
2) By reading the review the impression is generated that lipid nanoparticles can be divided into two groups, SLN and all others called NLC. Is that correct?
3) Lipid nanoparticles are used in mRNA vaccines and few lines are desirable to explain the differences between them and here reviewed SLN;
4) What time span is covered in this review? How does it complement the recent review by Borges et al. (SLNs as carriers of natural phenolic compounds, Antioxidants (2020))?
Author Response
Response to Reviewer 1 Comments
This is a review which might very useful for someone entering the field of lipid nanoparticles. Particularly well covered is the subfield of encapsulation of antioxidant compounds.
Some minor modifications are needed in order to improve the overall impression:
Point 1: There are no figures! I strongly recommend placing few illustrations to enhance the review readability;
Response 1:
Few illustrations have been added to the manuscript to enhance the review readability (Figure 1, Figure 2, Figure 3, and Figure 4: Page 4, Line 179-185, Pages 5, Line 199-206, Page 7, Line 253-257, and Page 10, Line 378-381, red color).
Point 2: By reading the review the impression is generated that lipid nanoparticles can be divided into two groups, SLN and all others called NLC. Is that correct?
Response 2:
Yes, that's correct; lipid nanoparticles in the form of solid lipids can be divided into two major types or groups, namely SLN and NLC as described in the review article.
Point 3: Lipid nanoparticles are used in mRNA vaccines and few lines are desirable to explain the differences between them and here reviewed SLN;
Response 3:
The use of lipid nanoparticles in mRNA vaccines and few lines have been explained in the manuscript (Pages 13, lines 474-487, and Table 4, Page 15, Lines 489-490, red color).
Point 4: What time span is covered in this review? How does it complement the recent review by Borges et al. (SLNs as carriers of natural phenolic compounds, Antioxidants (2020))?
Response 4:
The time span covered in this review is mainly from 2010 to January 2023. Borges et al., 2020 focused on phenolic compounds, while this review complements it by covering all antioxidants or other active compounds, both phenolic and non-phenolic, both polar and non-polar antioxidant compounds.

Reviewer 2 Report
Comments are in attached file
The document should focus only on SLNs, highlighting their properties and the effect of the lipids' characteristics.
Or otherwise perform a comparative analysis between SLNs and NLCs

Author Response
Response to Reviewer 2 Comments
Point 1: Authors compare The SLNs vs NLC then the title has been changed.
Response 1:
The SLN vs NLC is compared to find out some of the differences, advantages, and disadvantages before discussing more deeply about SLNs.
Point 2: Repetitive sentence, is necessary consider the relevant analysis of review.
Response 2:
The relevant analysis of the review has been considered and added to the text (Pages 1, lines 9-11, red color).
Point 3: Nothing is mentioned regarding the effect of crystal formation from lipid used, their control and the effect on the encapsulation efficiency.
Response 3:
The effect of crystal formation from lipids used, their control and the effect on the encapsulation efficiency have been mentioned (Pages 1, lines 17-20, red color).
Point 4: According from information analysis, which the recommended method to preparation of SLN.
Response 4:
The recommended method for the preparation of SLN has been added; namely, High-shear homogenization combined with ultrasonication is the recommended method and has been widely used because of the ease of preparation and good results. (Pages 1, lines 23-25, red color).
Point 5: Why the authors compared the SLNs and NLCs. The authors do not sohw the effect of temperature and preparation conditions on the functionality of SLNs.
Response 5:
SLNs and NLCs are compared because each type of lipid nanoparticle has advantages and disadvantages, as explained in the review. The effect of temperature and preparation conditions on the functionality of SLNs has been showed/added (Pages 3, lines 111-116, red color).
Point 6: Effect of the physical properties of lipid solid does not analysed.
Response 6:
Effect of the physical properties of lipid solid has been added (Page 4, lines 149-157, red color).
Point 7: Check the article: Solid Lipid Nanoparticles: An Approach to Improve Oral Drug Delivery.
Response 7:
The article: Solid Lipid Nanoparticles: An Approach to Improve has been checked and has been added to deepen the discussion (Pages 12, lines 438-443, red color).

Reviewer 3 Report
The manuscript entitled "Solid Lipid Nanoparticles: Review of The Current Research for Encapsulation and Delivery System of Active and Antioxidant Compounds" authored by Edy Subroto and co-workers is a comprehensive review article, overviewing state-of-the art of Solid Lipid Nanoparticles current research and developments in delivery of antioxidant and active molecules. The review is an interesting paper discussing\reviewing current state of antioxidant durg delivery research by using SLNs. It is surely suitable for Antioxidants Journal broad audience.However it needs some improvements and corrections of few minor issues, before resubmission in its final form. I have listed below my remarks\suggestions\changes for reaching standard for beeing publishable in Antioxidants
1) I would reccomend an improvement of Literature references by updating them with novel reviews or research articles
2) A thorough (although) moderate English revision by mother tongue is also suggsted for polishing language from few typos or imperfections in the text.
3) A more deep critical discussion in the final part of the article is also suggested for further improving the interest and \or comprehesion of the thematic
4) In the review are present only tables. I would strongly suggest to a add new Figures (taken from recent literature, eg SEM\TEM images of SLNs) and\or schemes of preparation and\or applications for SLNs and\or comparison with NLCs.
Author Response
Response to Reviewer 3 Comments
The manuscript entitled "Solid Lipid Nanoparticles: Review of The Current Research for Encapsulation and Delivery System of Active and Antioxidant Compounds" authored by Edy Subroto and co-workers is a comprehensive review article, overviewing state-of-the art of Solid Lipid Nanoparticles current research and developments in delivery of antioxidant and active molecules. The review is an interesting paper discussing\reviewing current state of antioxidant durg delivery research by using SLNs. It is surely suitable for Antioxidants Journal broad audience.However it needs some improvements and corrections of few minor issues, before resubmission in its final form. I have listed below my remarks\suggestions\changes for reaching standard for beeing publishable in Antioxidants
Point 1: I would reccomend an improvement of Literature references by updating them with novel reviews or research articles
Response 1:
Literature references have been improved by updating them with novel reviews or research articles (Pages 26, lines 775-776, Page 27, lines 813-814, 820-824, 846, Page 29, lines 899-900, Page 30, lines 939-941, 964-973, Page 31, lines 999-1001, Page 32, lines 1035-1036, 1039-1040, Page 33, lines 1085-1087, 1098-1102, red color).
Point 2: A thorough (although) moderate English revision by mother tongue is also suggsted for polishing language from few typos or imperfections in the text.
Response 2:
English language has been revised for polishing language from few typos or imperfections in the text.
Point 3: A more deep critical discussion in the final part of the article is also suggested for further improving the interest and \or comprehesion of the thematic
Response 3:
A more deep critical discussion in the final part of the article has been revised to improve the interest and \or comprehension of the thematic (Pages 19, lines 540-549, Pages 22, lines 624-632 Pages 23, lines 652-655, red color).
Point 4: In the review are present only tables. I would strongly suggest to a add new Figures (taken from recent literature, eg SEM\TEM images of SLNs) and\or schemes of preparation and\or applications for SLNs and\or comparison with NLCs.
Response 4:
New Figures (taken from recent literature, eg SEM\TEM images of SLNs) and\or schemes of preparation and\or applications for SLNs and\or comparison with NLCs have been added to the manuscript (Figure 1, Figure 2, Figure 3, and Figure 4: Page 4, lines 179-185, Pages 5, lines 199-206, Page 7, lines 253-257, and Page 10, lines 378-381, red color).

Round 2
Reviewer 1 Report
The authors improved the text substantially after taking into account not only mine but also the comments of other two reviwers.
I can recommend the work for publication in the present form.
Reviewer 2 Report
No comments